# Natural Catalytic IgGs Hydrolyzing Histones in Schizophrenia: Are They the Link between Humoral Immunity and Inflammation?

**DOI:** 10.3390/ijms21197238

**Published:** 2020-09-30

**Authors:** Evgeny A. Ermakov, Daria A. Parshukova, Georgy A. Nevinsky, Valentina N. Buneva

**Affiliations:** 1Institute of Chemical Biology and Fundamental Medicine, Siberian Branch of the Russian Academy of Sciences, 8 Lavrentiev Ave., 630090 Novosibirsk, Russia; evgeny_ermakov@mail.ru; 2Department of Natural Sciences, Novosibirsk State University, 2 Pirogova St., 630090 Novosibirsk, Russia; 3Mental Health Research Institute, Tomsk National Research Medical Center of the Russian Academy of Sciences, 634014 Tomsk, Russia; susl2008@yandex.ru

**Keywords:** schizophrenia, histones, DAMPs, antibodies, IgG proteolytic activity, abzymes, humoral immunity

## Abstract

Schizophrenia is known to be accompanied not only with an imbalance in the neurotransmitter systems but also with immune system dysregulation and chronic low-grade inflammation. Extracellular histones and nucleosomes as damage-associated molecular patterns (DAMPs) trigger systemic inflammatory and toxic reactions by activating Toll-like receptors. In this work, we obtained the first evidence that polyclonal IgGs of patients with schizophrenia effectively hydrolyze five histones (H1, H2a, H2b, H3, and H4). Several strict criteria were used to demonstrate that histone-hydrolyzing activity is a property of the analyzed IgGs. The IgGs histone-hydrolyzing activity level, depending on the type of histone (H1–H4), was statistically significantly 6.1–20.2 times higher than that of conditionally healthy donors. The investigated biochemical properties (pH and metal ion dependences, kinetic characteristics) of these natural catalytic IgGs differed markedly from canonical proteases. It was previously established that the generation of natural catalytic antibodies is an early and clear sign of impaired humoral immunity. One cannot, however, exclude that histone-hydrolyzing antibodies may play a positive role in schizophrenia pathogenesis because histone removal from circulation or the inflamed area minimizes the inflammatory responses. Thus, it can be assumed that histone-hydrolyzing antibodies are a link between humoral immunity and inflammatory responses in schizophrenia.

## 1. Introduction

Histones and their post-translational modifications play a key role in chromatin remodeling and gene transcription. In addition to intranuclear functions, histones act as damage-associated molecular patterns (DAMPs) [1,2] when they are released into the extracellular space, exhibiting significant toxic or inflammatory activity in vivo and in vitro [3,4]. The administration of exogenous histones to laboratory animals leads to systemic inflammatory reactions through the activation of Toll-like receptors and downstream inflammatory pathways. Anti-histone treatment (for instance, specific neutralizing antibodies, recombinant thrombomodulin, heparin, and activated protein C) protect animals from sepsis and lethal endotoxemia, ischemia/reperfusion injury, stroke, peritonitis, pancreatitis, thrombosis and hypercoagulation [5]. Besides, elevated levels of serum histones and nucleosomes are involved in multiple pathophysiological processes and disease progression, including inflammatory and autoimmune diseases (AIDs) and even oncological pathologies [5,6,7]. Therefore, extracellular histones can be considered as biomarkers and new promising therapeutic targets in several human diseases.

Histones and nucleosomes can enter the extracellular space and the bloodstream as a result of apoptosis, pyroptosis, necrosis, and other programmed cell death variants [5]. In some cases, cell death can play a protective role. For example, during NETosis neutrophils release extracellular traps (NETs) containing DNA and histones (from nuclear or mitochondrial chromatin), which act not only as a physical barrier but also exhibit bactericidal activity, neutralizing various pathogens [8,9]. However, at excessive concentrations, extracellular histones and nucleosomes lead to systemic inflammatory and toxic reactions, contributing to the development of autoimmune responses [5,6]. Therefore, the removal of circulating extracellular histones can reduce inflammatory and autoimmune responses.

Dysfunctional apoptosis can lead to the appearance of cell-free DNA (cfDNA) in the bloodstream. It is shown that cfDNA levels are significantly elevated in blood of patients with schizophrenia [10,11]. Besides, the activation of apoptosis in schizophrenia has been shown using the analysis of proteins associated with apoptosis, as well as by measuring the activity of caspase-3 in both peripheral blood and the brain [10]. The accumulation of fragments of cfDNA in the blood promotes inflammatory responses [10,11] as well as the recognition of cfDNA as an antigen and the generation of autoantibodies.

A plethora of evidence indicates the dysregulation of both innate and adaptive immune systems in schizophrenia [12,13,14,15]. Evidence includes increased levels of pro-inflammatory cytokines, genetic predisposition, transcriptome changes, epidemiological data and positive effects of anti-inflammatory therapy [12,13,14,15]. In our previous studies, we revealed new evidence of an impaired immune system in schizophrenia. A significant increase in the titers of antibodies to double-stranded (ds) DNA in serum of patients as compared with healthy donors was revealed [16,17]. Observations of an increased titer of anti-DNA antibodies in schizophrenia are consistent with previously published data [18,19]. It is well known that autoantibodies to dsDNA are one of the biomarkers of systemic lupus erythematosus (SLE) [20,21]. Moreover, we detected immunoglobulins containing a light chain with reduced molecular weight (17.3 kDa), similar to those for SLE [17]. The obtained data on the increase in the titers of antibodies to dsDNA in schizophrenia may indicate a violation of immunological homeostasis, similar to that in SLE. It is important to note that from 14% to 75% of patients with SLE demonstrate neuropsychiatric symptoms, including mood and behavior disorders, as well as psychotic symptoms [22,23], which may also indicate a link between immunological disturbances and the development of psychopathology.

Anti-DNA antibodies are usually directed against nucleosomal histone–DNA complexes [24,25]. Antinuclear antibodies are known to be formed upon immunization with histone–DNA complexes, nucleosomes, or histones [24,25]. The above observations on the presence of anti-DNA autoantibodies in schizophrenia suggest that limited subgroups of patients exhibit more pronounced disturbances in the immune system.

Previously, in several laboratories, it was shown that the activation of the immune system accompanies the formation of catalytic antibodies or abzymes [26,27,28,29,30,31,32,33,34]. These natural catalytic antibodies are capable of not only binding antigens but also hydrolyzing them [26,27,28,29,30,31,32,33,34]. Antigen hydrolysis is one example of the non-canonical functions of immunoglobulins and reflects the extreme diversity of types of antibodies generated by B cells [35,36]. Using bioinformatics, biochemical and crystallographic methods, it has been shown that the complementarity-determining regions (CDR) of some antibodies contain characteristic motifs (catalytic triad) typical of classical proteases [35,36]. However, the detailed molecular mechanisms for the synthesis of catalytic antibodies by plasma cells are still not fully understood.

Natural catalytic antibodies hydrolyzing DNA, RNA, polysaccharides, and various proteins have been detected in some AIDs and viral diseases [26,27,28,29,30,31,32,33,34,36]. It is noteworthy that specific fractions of catalytic antibodies appear in the human antibody repertoire long before significant changes in the titers of autoantibodies and the onset of clinical symptoms of certain diseases [30,31,32,33,34]. It was shown that catalytic antibodies’ generation is the earliest and most substantial sign of impaired humoral immunity. Recently, we found catalytic antibodies with DNase and RNase activities, recognizing DNA and RNA and hydrolyzing non-specific homo-oligonucleotides, and site-specific microRNAs in patients with schizophrenia [17,37,38]. These observations are consistent with previously published data and indicate the formation of antinuclear antibodies directed against nucleic acids and nuclear proteins, including histones in schizophrenia [19].

Antibodies hydrolyzing histones were recently detected in multiple sclerosis (MS) [39,40], SLE [41], and HIV infection [42,43,44,45]. Catalytic antibodies against histones in HIV infection hydrolyze them at sites localized in the protein sequences of histones interacting with DNA and their antigenic determinants [42,43,44,45]. Histones-hydrolyzing abzymes can probably play a protective role by participating in the removal of histones from the bloodstream.

The literature describes the widespread phenomenon of cross-complexing certain antibodies and enzymes with foreign ligands-antigens [32,33,34,46]. At the same time, cross-catalysis in the case of canonical enzymes is an extremely rare occurrence [47]. It was shown that abzymes against any specific protein usually hydrolyze only this, but no other control proteins [30,31,32,33,34]. The first examples of cross-catalytic activity of antibodies have been discovered recently [41,47]. It has been shown that IgG antibodies against human histones effectively hydrolyze myelin basic protein (MBP) and, conversely, antibodies against MBP cleave histones [41,47]. Therefore, catalytic cross-reactive antibodies against histones and MBP can play a negative role, contributing to the development of the disease [47]. Recently, we detected catalytic antibodies in serum of patients with schizophrenia that hydrolyze MBP [48]. Therefore, it was interesting to establish whether histone-hydrolyzing antibodies are formed in schizophrenia.

Here, we investigated the ability of IgGs from patients with schizophrenia to hydrolyze various histones, and also analyzed their catalytic properties in detail: dependence on pH and metal ions, kinetic characteristics. A correlation of IgG’s relative activity in the hydrolysis of some histones with clinical parameters of schizophrenia has also been analyzed.

## 2. Results

### 2.1. Characterization of Schizophrenia Patients and Healthy Donors Participating in the Study

In this work, we recruited 50 schizophrenia patients and 16 conditionally healthy donors. Among 50 schizophrenia patients, 22 patients had prevailing positive symptoms, and 28 patients had prevailing negative symptoms. According to ICD-10 classification, 11 patients were diagnosed with F20.00 (paranoid schizophrenia with continuous course), 14 patients with F20.01 (paranoid schizophrenia; episodic with the progressive deficit), 11 patients with F20.02 (paranoid schizophrenia; episodic with the stable deficit), eight patients were diagnosed with F20.09 (paranoid schizophrenia; course uncertain), and six patients were diagnosed with F20.6 (simple schizophrenia). All study participants had no signs of active inflammatory or autoimmune diseases, tumors, temporal lobe epilepsy, or encephalitis that contribute to the development of organic psychosis. More detailed characteristics of patients and healthy donors are presented in the Section 4.2 and the Appendix A.

### 2.2. IgGs Purification and Evidence of Their Histone Hydrolyzing Activity

IgG preparations were isolated from the sera preparations of 50 schizophrenia patients and 16 healthy donors using protein-G Sepharose affinity chromatography according to our previously developed protocol providing electrophoretically homogeneous antibodies [26,35,36,46]. To check the homogeneity of the obtained IgGs, we used SDS-PAGE analysis with silver staining. Figure 1A presents the analysis of IgG_mix_ preparation (an equimolar mixture of the ten most active IgG preparations). One band in the initial intact IgG_mix_ (150 kDa, lane 1) was revealed. Incubation of IgG_mix_ with dithiothreitol (DTT) at 37 °C (conditions for the incomplete reduction of disulfide bonds in IgG molecules) caused the formation of three bands (lane 2): 100 kDa (HH-dimer), 50 kDa (H-chain) and 25 kDa (L-chain). After incubation of IgG_mix_ with DTT at 100 °C (conditions for the complete reduction of disulfide bonds), only two bands (lane 3) were revealed: 50 kDa, H-chain and 25 kDa, L-chain. It should be noted that lane 3 has less protein applied, so it has a lower intensity. Nonetheless, no additional bands were identified. Thus, the obtained IgGs did not contain any visible impurities of other proteins.

The analysis of histones-hydrolyzing activity was carried out using SDS-PAGE. As shown in Figure 1B, the incubation of a histone mixture (H1, H2a, H2b, H3, and H4) with IgG preparations of patients with schizophrenia led to a decrease in the intensities of Coomassie-stained initial histones bands on a gel and the appearance of their various fragments with a lower molecular mass. IgG preparations from healthy donors did not exhibit visible hydrolysis histones.

Then, IgGs preparations were subjected by fast protein liquid chromatography (FPLC) gel filtration of IgG_mix_ (Figure 2) under acidic conditions (pH 2.6) with subsequent determination of the activity in the obtained fractions as in [26,35,36,46] to prove that histones-hydrolyzing activity belongs directly to IgGs. Under these conditions, the destruction and separation of all non-covalent protein complexes usually occurred. The coincidence of the gel filtration profiles (A_280_) and profiles of the proteolytic activity of IgG_mix_ in the hydrolysis of five histones and the absence of any other peaks (Figure 2) provides reliable evidence that IgGs possessed histone-hydrolyzing activity and did not contain any impurities of other proteases.

### 2.3. Comparison of IgGs Histone-Hydrolyzing Activity of Healthy Individuals and Patients

According to the screening results, the level of the histone-hydrolyzing activity of IgGs varied widely (Figure 3A). The level of relative IgGs proteolytic activity was determined by decreasing the intensity of each histone’s protein bands due to hydrolysis (complete hydrolysis of the substrate was taken as 100%) (see Materials and Methods section and Figure 1B). Interestingly, the hydrolysis level of five histones by IgGs of patients with schizophrenia was statistically significantly (*p* < 0.0006) 6.1–20.2 times higher than by antibodies from healthy individuals (Figure 3). Depending on the histone, the differences were as follows (-fold): H1 (7.3), H2a (8.2), H2b (6.5), H3 (6.1), and H4 (20.2). However, the activity of IgG preparations of patients is heterogeneous; for example, eight cases are identified using Tukey’s test as outlying for histone H1. Besides, among 50 IgGs of schizophrenia patients, 34 preparations (68%) hydrolyzed all five histones, 14 antibodies (28%) split from 1 to 4 histones, and only two IgGs (4%) did not hydrolyze a single histone (Figure 3B). At the same time, among all IgGs preparations of patients, ten preparations had a high level of activity (>20% substrate hydrolysis) in the hydrolysis of all five histones, of which seven IgGs preparations of patients with paranoid schizophrenia demonstrated the highest activity (>40% substrate hydrolysis). In contrast, among 16 IgGs of healthy donors, only three preparations (19%) hydrolyzed with low intensity all five histones, nine antibodies (56%) weakly cleaved 1–4 histones, and four preparations (25%) did not hydrolyze a single histone (Figure 3B). However, it should be emphasized that IgGs of healthy donors had a lower level of activity than IgGs of schizophrenia patients (see Figure 3A).

The following analysis of the relative IgGs proteolytic activity, depending on the prevailing symptomatology, showed that the average values of histone-hydrolyzing activity of IgGs of schizophrenia patients with leading negative symptoms were 1.2–1.9-fold higher than that of patients with leading positive symptoms (Table 1). However, these differences were statistically insignificant (in all cases, *p* > 0.05). That is also indicated by the fact that the median values of the IgGs histone-hydrolyzing activity levels were very close in all compared groups (Table 1). In addition, the average values of the hydrolysis levels of all five histones (H1–H4) by IgG preparations in the general group of patients slightly differed from the average values in the groups of patients with leading positive and negative symptoms, and the median values in all analyzed groups were comparable.

We analyzed the levels of histone-hydrolyzing activity of IgGs of patients depending on the type of schizophrenia in accordance with the International Classification of Diseases (ICD-10) (Table 2). Although the average values of IgGs proteolytic activity of patients with various types of schizophrenia differed in some cases by more than two times, nevertheless, the median values of the levels of IgGs activities were comparable. Using the Kruskal–Wallis one-way analysis of variance (ANOVA), we did not find significant differences (*p* > 0.05) between the levels of histone-hydrolyzing activity of IgGs of different patients depending on the type of schizophrenia.

Correlation analysis (Table 3) showed that the hydrolysis levels of various histones (H1–H4) by IgGs of schizophrenic patients were clearly positively correlated with each other (correlation coefficients *r* > 0.6, *p* < 0.05). Thus, IgGs from schizophrenic patients hydrolyzed various histones with comparable efficacy.

No statistically significant correlation was found between IgGs proteolytic activity levels, either with the duration of the disease or with the age of the patients (Table 3). However, it is essential to note that a positive correlation was found between IgGs relative activity in the hydrolysis of histone H2b and H4 with scores according to the general symptoms subscale of the positive and negative syndrome scale (PANSS) (*r* = 0.288 and *r* = 0.314, *p* < 0.05, respectively) [49]. The correlation coefficients between the hydrolysis level of histones H1, H3, and H2a and scores on the PANSS general symptoms scale were low and statistically insignificant (ranged from 0.136 to 0.264). Besides, other correlations between the IgGs activity levels and scores on the PANSS scales were not found, including the positive syndromes scale (PANSS Positive) and negative syndromes scale (PANSS Negative) (Table 3). These data are indirectly consistent with data on the absence of significant differences between the IgGs histone-hydrolyzing activity levels in patients with positive and negative symptoms (Table 1).

Since the nuclease activity of IgGs from patients used in this analysis was studied earlier [26,35,36], it was possible to conduct a retrospective correlation analysis (Table 4). The correlation coefficients of the histone-hydrolyzing activity of IgGs, with the level of DNase activity in the hydrolysis of plasmid DNA and RNase activity in the hydrolysis of various microRNAs (miR-219a-5p, miR-219a-2-3p, miR-9-5p, miR-137), were estimated. Significant positive correlation coefficients were found between the IgGs relative activities in the hydrolysis of histones H1, H2a, and H4 with DNase activity of IgGs (*r* > 0.4, *p* < 0.05). Besides, significant positive correlation coefficients were found between the hydrolysis of histones H1 and H4 and the level of RNase activity in the hydrolysis of miR-219a-5p and miR-219a-2-3p (*r* > 0.4, *p* < 0.05). However, no correlation of IgGs protease activity levels with the IgGs RNase activity levels in the hydrolysis of miR-9-5p and miR-137 was identified.

### 2.4. The Catalytic Properties of Histone-Hydrolyzing IgGs in Schizophrenia

Analysis of the dependence of the histone-hydrolyzing activity level of an equimolar mixture of nine high-activity schizophrenia IgGs (IgG_mix_) on the pH of the reaction medium (Figure 4) showed that the antibodies of patients effectively hydrolyzed all five histones in the entire pH range, but the highest catalytic activity was observed at pH 5.5. Besides, for histone H1, there is a pronounced increase in the IgGs activity at pH 7.5. It is important to note that the efficiency of histone hydrolysis by IgG_mix_ at various pH varies depending on the hydrolyzable histone. For example, the hydrolysis of histones H2a and H2b at pH 8.5 is the lowest, while the hydrolysis of histones H1 and H3 is much higher at the same pH.

An analysis of the metal dependence of the histone-hydrolyzing activity of IgG_mix_ showed that its dialysis against buffer with ethylenediaminetetraacetic acid (EDTA) leads to the inhibition of hydrolysis of all five histones by more than 60% (Figure 5). The addition of external Ca^2+^ ions led to a significant increase in IgGs catalytic activity in all histones’ hydrolysis by dialyzed IgG_mix_. Moreover, this activity increased above the activity level of intact antibodies. Other metal ions slightly affected the activity, and the effect depended on the hydrolyzable histone. In some cases, metal ions showed an inhibitory effect, in others—slightly activating. However, the most pronounced activating effect was demonstrated by the Ca^2+^ ion (Figure 5).

Analysis of substrate specificity demonstrated that the IgGs of two different schizophrenia patients effectively hydrolyzed five histones and MBP, but did not hydrolyze other control proteins (human and bovine serum albumin, human milk lactoferrin, human lysozyme) under the same reaction conditions (Figure 6A,B). Hydrolysis of MBP by antibodies from patients with schizophrenia has already been shown [46]. However, in this article, the polyclonal IgGs were not subjected to affinity chromatography on histone-Sepharose and MBP-Sepharose for obtaining specific IgGs against these proteins. Thus, it should be assumed that histones are hydrolyzed mainly by anti-histones antibodies, while MBP is by anti-MBP antibodies. However, it cannot be excluded, as in the case of IgGs from the sera of SLE or HIV-infected patients [39,45], that these antibodies of schizophrenia patients can also possess cross-catalytic activity.

Catalytic antibodies can bind antigen with different affinities, which determine the rate of catalysis. To estimate the kinetic parameters of histone hydrolysis influenced by IgG, the Michaelis–Menten approach and Lineweaver–Burk coordinates were used (see Materials and Methods section). The dependences of the histone hydrolysis rate on the substrate concentration for two IgG preparations confirmed the Michaelis–Menten kinetics (Figure 7). Similar clinical data characterize two patients with a paranoid form of schizophrenia. They differed only in gender, duration of the disease, type of pathology course: IgG-1 (male, 9 years of illness, paranoid schizophrenia with episodic with a progressive deficit (F20.01)), and IgG-2 (female, paranoid schizophrenia with a continuous course (F20.00)). IgGs of these patients demonstrated comparable proteolytic activity. However, these two IgGs demonstrate different kinetic parameters. The apparent *K*_m_ values in the hydrolysis of five histones by IgG-1 and IgG-2 varied from 2.3 to 24.1 μM, while *k*_cat_—from 3.9 × 10^3^ to 11.1 × 10^3^ min^−1^ (Table 5). The highest affinity was observed for the IgG-1 preparation in the hydrolysis of histones H3 and H1: *K*_m_ = 2.3 and 2.8 μM, respectively. The lowest affinity show IgG-2 for H2a (*K*_m_ = 8.9 μM) and H4 (*K*_m_ = 24.1 μM) (Table 5). Thus, during the schizophrenia development, abzymes can be produced against five histones with very different affinities and relative activity in their hydrolysis.

## 3. Discussion

In this work, we discovered antibodies hydrolyzing histones from the serum of patients with schizophrenia and described their biochemical properties. Earlier, we developed a method protocol for the isolation of electrophoretically homogeneous antibodies from different biological fluids that do not contain impurities of any canonical enzymes, which was used in numerous studies [17,30,31,32,33,34,37,38,39,40,41,42,43,44,45,47,48,50]. Fourteen previously proposed strict criteria were used to show that antibodies obtained using this method [26,30,31,32,33,34] do not contain impurities of any classical enzymes [30,31,32,33,34]. In this work, electrophoretically homogeneous IgGs were obtained using this protocol. Besides, we have demonstrated the fulfillment of the following criteria, which reliably confirm that antibodies possess histone-hydrolyzing activity: (1) electrophoretic homogeneity of the analyzed antibodies (using silver staining of protein) (Figure 1A); (2) the presence of high activity of antibodies from the blood of patients, while antibodies from the blood of healthy individuals possessed low activity (Figure 3); and (3) after gel FPLC filtration of antibodies under conditions of the disintegration of non-covalent complexes (pH 2.6), the profile of IgG gel filtration coincides with the profile of proteolytic activity in the obtained fractions (Figure 2).

The generation of antibodies with catalytic activity is associated with the formation of autoantibodies. In schizophrenia, dysfunction of apoptotic cell death is found, which leads to the release of cfDNA, and also nucleoproteins including histones, into the bloodstream [10,11]. This can lead not only to inflammatory responses after recognition of cfDNA and histones by Toll-like receptors [5,11], but also to their presentation as antigens and the generation of autoantibodies. This assumption is supported by the fact that there is an increase in anti-DNA antibody titers in schizophrenia [16,17,18,19,51]. Anti-DNA antibodies are also known to cross-bind to the nucleosome and histone proteins [24,25]. It can be assumed that IgGs with histone-hydrolyzing activity are present among generated anti-DNA and anti-histone autoantibodies. Thus, one of the reasons for the generation of histone-hydrolyzing IgGs in schizophrenia may be an increased level of cfDNA and histones in the bloodstream due to dysfunction of apoptosis.

Screening of the histone-hydrolyzing activity level of antibodies from the serum of schizophrenic patients statistically reliably showed its 6.1–20.2-fold higher level in comparison with the level of healthy donors (*p* < 0.0006) (Figure 3A). However, it should be noted that only 20% of IgGs of patients with schizophrenia had high activity (>20% substrate hydrolysis). These data indicate that these patients have more anti-histone antibodies. These data are consistent with the available data that 23% of schizophrenic patients have anti-histone IgG, while only 6.6% of healthy subjects were positive for anti-histone IgG [52]. Besides, it was previously demonstrated that catalytic antibodies’ generation is associated with immunological disturbances that appear long before the onset of typical symptoms of autoimmune pathologies [32,33,34,36]. Therefore, these data indicate pronounced disturbances in humoral immunity in schizophrenia, at least in some subgroups of patients (with a high level of IgG proteolytic activity). Therefore, the data obtained can be used to identify patients with severe disturbances of humoral immunity.

There was no statistically reliable difference in the average values in the hydrolysis of individual histones by IgGs of schizophrenia patients with leading negative and leading positive symptoms (Table 1). Nevertheless, the obtained data may indicate a tendency to increase 1.5-fold the level of histone hydrolyzing activity of IgG in patients with negative symptoms. A similar relationship was observed for the MBP-hydrolyzing activity of IgG in schizophrenia [48]. The obtained results suggest that the histone-hydrolyzing activity of IgGs can depend on other features of the disease.

Based on the analysis of histone-hydrolyzing activity levels of IgGs depending on the type of schizophrenia (Table 2), it can be assumed that the level of the activity depends on the degree of disturbance in the patients’ humoral immune system but does not remarkably depend on the type of schizophrenia. However, data on the catalytic activity of IgG in schizophrenia, along with other immunological parameters, can be used to stratify the patients according to the degree of immune system disturbances. The identification of patients with severe immune disorders will allow them to prescribe anti-inflammatory and immunomodulatory therapy purposefully. 

In this work, a positive correlation was found between the hydrolysis level of histones H2b and H4 by IgGs of patients with scores on the PANSS general symptoms subscale (Table 3). It may indicate that the level of the histone-hydrolyzing activity of IgGs may reflect the general condition of the patient. Besides, a significant positive correlation in the hydrolysis level between different histones is shown (Table 3). These findings indicate that the mechanisms of generation of natural catalytic IgG hydrolyzing histones in schizophrenia are similar. Nevertheless, it can be assumed that some subfractions of catalytic antibodies may be formed for each individual histone, and the mechanisms of formation of abzymes for diverse antigens may differ.

According to the results of a retrospective analysis for the same samples of schizophrenia IgGs, a clear positive correlation was found between the levels of hydrolysis of H1 and H4 with that for RNase activity (hydrolysis of miR-219-2-3p and miR-219-5p) [37,38], as well as activity in the hydrolysis of H1, H2a, and H4 with DNA-splitting activity [17] (Table 4). Anti-DNA antibodies are known to generate against complexes of nucleic acids with various proteins, particularly to complexes of DNA with histones [24,25]. These data of correlation analysis indirectly confirm our assumption that histone-hydrolyzing antibodies are formed among the pool of anti-histone antibodies, due to the availability of cfDNA and histones as an antigen.

This work revealed that the catalytic properties of histone-hydrolyzing IgG in schizophrenia significantly differed from canonical serum proteases. Surprisingly, the highest proteolytic activity of IgG preparations was at pH 5.5 (Figure 4). These data indicate that part of the studied catalytic IgGs exhibit a type of catalysis characteristic of acidic proteases, for instance, cathepsins—intracellular cysteine proteases, for which the optimum pH lies in the range of pH 4–6.8 [53]. It is known that cathepsins, along with caspases, are associated with the regulation of cell death [54,55,56] and other different physiological and pathological processes, such as maturation of the MHC class II complex, bone remodeling, keratinocyte differentiation, tumor progression, and metastasis, rheumatoid arthritis, and osteoarthritis, and also atherosclerosis [57]. Cathepsins are mainly found in lysosomes and can only leave the cell under certain physiological and pathological conditions [56,57], whereas immunoglobulins are found extracellularly or circulate in the bloodstream. The area of inflammation is known to have an acidic pH. Therefore, it is possible that natural catalytic IgGs, which have an “acidic” pH optimum, can participate in the destruction of histones that are released from dying cells in areas of inflammation, thereby reducing inflammatory responses.

The activating effect of Ca^2+^ ions on the histone-hydrolyzing activity of antibodies in the hydrolysis of all five histones turned out to be rather unusual (Figure 5). In the case of the histone-hydrolyzing IgGs of HIV-infected patients, only one IgG preparation of the patient demonstrated a similar activating effect of Ca^2+^ ions. In contrast, IgGs preparations of other patients were activated by other different metal ions [42]. The Ca^2+^-dependent activation of histones’ hydrolysis by IgGs in the case of MS patients was very low; these antibodies were activated significantly better by Al^3+^ and Fe^2+^ ions [39]. However, the same activating effect of Ca^2+^ ions was found in the hydrolysis of MBP by IgGs of schizophrenia patients [48], which suggests that Ca^2+^-dependent proteolytic antibodies to different proteins are formed in schizophrenia, and also that similar mechanisms generate them. Therefore, it can be proposed that the generation of Ca^2+^-dependent IgGs is a specific feature of schizophrenia. Canonical Ca^2+^-dependent proteases are calpains involved in inflammatory reactions in vascular endothelial cells and are expressed in brain endothelial cells [58]. Consequently, it can be assumed that fractions of Ca^2+^-dependent catalytic IgGs have a similar mechanism of catalysis with calpains or may be second anti-idiotypic antibodies to active centers of calpains.

Analysis of the kinetic parameters of the antibody-catalyzed histone hydrolysis reaction showed that the *K*_m_ values for the histone H1 hydrolysis (2.8–4.8 μM) are comparable with those for IgG of MS patients cleaving histone H1 (4.0–6.7 μM) [39] and for IgGs of HIV-infected patients (1.0–5.6 μM) [42], which indicates comparable affinity of IgGs of patients with schizophrenia compared to IgGs of patients with HIV infection and MS. The *K*_m_ values reflect the affinity of the enzyme to the substrate (in our case, antibodies to the antigen-substrate). Based on the data obtained, it follows that IgGs of patients with schizophrenia have a higher affinity than canonical proteases. However, it should be noted that pathogenic tight binding autoantibodies usually have a high affinity (K_d_ range: 10^−6^–10^−10^ M) [59]. Therefore, some antibodies in schizophrenia acquire catalytic activity and rather high rates of catalysis due to their lower affinity than those of tight binding autoantibodies. Binding affinity is determined by the number of bonds formed with the antigen, and accordingly is determined by the amino acid residues that form the CDR [60]. Therefore, it can be assumed that the antibodies of patients with catalytic activity have a specific CDR structure.

It was previously shown that the formation of catalytic antibodies is associated with an impaired differentiation profile of hematopoietic stem cells in the bone marrow using animal models of autoimmune diseases [61,62]. Therefore, based on the data obtained in this work, it can be assumed that immunological disturbances in schizophrenia, including the formation of proteolytic IgG, are associated with impaired differentiation of bone marrow hematopoietic stem cells. One cannot exclude that such disturbances may be related to the effect of therapy in some cases. For example, it has been shown that atypical antipsychotics can affect hematopoietic stem cells and lead to an initial mobilization of CD34^+^ stem and progenitor cells into the circulation [63]. Additionally, indirect evidence of a possible association of schizophrenia with impaired hematopoietic stem cell differentiation may be because bone marrow transplantation leads to a remarkable reduction in psychopathologic symptoms without the administration of antipsychotic drugs [64]. However, the detailed mechanisms of the disorder’s development, leading to the synthesis of catalytic antibodies by B cells, have not yet been studied. Further research should be aimed at clarifying these mechanisms.

The obtained data indicate that histone-hydrolyzing IgGs can play both a positive and negative role in schizophrenia development. A positive role may be associated with the destruction and removal of histones from the bloodstream or inflammation areas. Since extracellular histones are DAMPs and induce inflammatory and autoimmune reactions [3,4,5], removing histones can reduce these reactions. However, the negative role of histone-hydrolyzing IgGs in schizophrenia may be associated with the catalytic cross-reactivity of anti-histone IgGs to MBP, similar to that for such antibodies in SLE and HIV-infected patients [41,47]. One cannot exclude that such enzymatic cross-reactivity of IgGs can contribute to a variety of autoimmune responses associated with the production of abzymes hydrolyzing histones and MBP. Besides, such catalytic cross-reactivity of IgGs may contribute to the destruction of the myelin sheath of nerve fibers and hypomyelination observed in schizophrenia [65]. Schizophrenia is known to be associated with chronic, low-grade inflammation [12,13,14,15]. Therefore, histone-hydrolyzing antibodies may be considered a new link between humoral immunity and inflammatory responses in schizophrenia.

## 4. Materials and Methods 

### 4.1. Materials and Reagents

In this work, many reagents and chemicals (human and bovine serum albumin, human milk lactoferrin, and human lysozyme) were parched from Sigma (St. Louis, MO, USA). Histones H1, H2a, H2b, H3, and H4 of calf thymus were from Sigma (product number 10223565001, St. Louis, MO, USA). MBP was from the Department of Biotechnology, Research Center of Molecular Diagnostics and Therapy (Moscow, Russia). Various reagents: NaCl, CaCl_2_, MgCl_2_, MnCl_2_, CuCl_2_, CoCl_2_, FeSO_4_, NiSO_4_, ZnSO_4_, 2-Amino-2- (hydroxymethyl) propane-1,3-diol (Tris), 2-morpholin-4-ylethanesulfonic acid (MES), 3-morpholinopropane-1-sulfonic acid (MOPS), Triton X-100, ethylenediaminetetraacetic acid (EDTA), dithiothreitol (DTT) were from MP Biomedicals (Eschwege, Germany). Protein-G-Sepharose and Superdex 200 HR 10/30 chromatographic columns were from GE Healthcare (GE Healthcare, New York, NY, USA).

### 4.2. Patients, Healthy Donors, and Biological Material

The bioethics committee of the Mental Health Research Institute (MHRI) of the Tomsk National Research Medical Center (TNRMC; Tomsk, Russia) reviewed and approved the protocol of this study, including the written consent of patients and healthy donors to use their blood for scientific purposes according to the Helsinki Declaration (protocol N 78/1.2015). All patients and healthy volunteers have signed Informed Consent to give their blood for this study. All schizophrenia patients were legally competent, so the consent of relatives or guardians was not required.

For the study, 50 schizophrenia patients (29 women and 21 men; mean age = 38.8 ± 10.4 years; disease duration = 13.8 ± 8.4) and 16 healthy donors (7 women and 9 men; mean age = 35.5 ± 8.2 years) were recruited during the period from January 2018 to December 2019. Among the patients included in the study, there were no first episode patients. The patients did not adhere to therapy and thus they were hospitalized in an acute state in the Department of Endogenous Disorders of the MHRI. Patients did not receive antipsychotic therapy for at least 1–6 months before hospitalization. Besides, prior to the study, neither healthy donors nor patients had taken drugs that affect the immune system, such as non-steroidal or steroidal anti-inflammatory drugs. Blood sampling was carried out before prescribing any treatment. Psychiatrists from the MHRI TNRMC, through structured clinical interviews, diagnosed the schizophrenia according to the Diagnostic and Statistical Manual of mental disorders IV (DSM-IV) and the International Statistical Classification of the Diseases and Related Health Problems (ICD-10) criteria, with symptom assessment using the positive and negative syndrome scale (PANSS) for schizophrenia [49]. Inclusion criteria were the following: paranoid or simple schizophrenia according to the International Statistical Classification of Diseases and Related Health Problems, 10th Revision (ICD-10: F20.0 and F20.6). Exclusion criteria for all study participants were the presence of any chronic infections, inflammatory and autoimmune diseases, and acute viral or bacterial infections no less than two months before the investigation, neurological diseases, other organic mental disorders, mental retardation, and drug addiction. Exclusion criteria for the control group were the presence of acute and chronic infectious, inflammatory, autoimmune, or neurological diseases, and organic brain disorders as well as episodes of drug use. Thus, participants with symptoms of active inflammatory or autoimmune diseases, tumors of the central nervous system, temporal lobe epilepsy, or encephalitis, that contribute to the development of organic psychosis, were not included in the study. Additional clinical data of patients are given in the Appendix A.

Vacuette tubes with coagulation activators were used for fasting venous blood sampling. Tubes with coagulated blood were centrifuged at 2000× *g* for 20 min in a Digicen 21 R refrigerated centrifuge (Orto Alresa, Madrid, Spain) to obtain serum preparations. Then the obtained serum was aliquoted and stored at −80 °C.

### 4.3. IgGs Purification from Human Serum

IgG preparations were obtained using affinity chromatography on the ÄKTA Start chromatograph (GE Healthcare) similar to previously published works [17,37,38,48,50]. The blood serum of patients and healthy donors (2 mL) was diluted four times with buffer A, consisting of 50 mM Tris-HCl, pH 7.5, and 150 mM NaCl. Then, it was applied to a Protein-G-Sepharose column (1 or 5 mL), previously equilibrated with the same buffer. Proteins that do not interact with the sorbent were washed with buffer A until zero optical density (A_280_). Nonspecifically bound proteins were eluted with buffer A containing 1% Triton X-100; the column was washed with buffer A until zero optical density. IgGs were explicitly eluted with 100 mM Gly-HCl buffer, pH 2.6. IgG preparations were further concentrated and additionally purified by FPLC gel filtration on a Superdex 200 HR 10/30 column. The obtained antibody fractions immediately after elution from the column were neutralized with 1 M Tris-HCl buffer, pH 8.8, and dialyzed against the buffer necessary for further work. To check the homogeneity of the obtained IgGs preparations, we used gradient 4–18% SDS-PAGE analysis with silver staining [66]. DTT reduction of disulfide bonds was used to separate IgG molecules into heavy (H) and light (L) chains. 

### 4.4. IgGs Histone-Hydrolyzing Proteolytic Activity Assay

An equimolar mixture of total histones (H1, H2a, H2b, H3, and H4) was used to determine the proteolytic activity of IgGs. The reaction mixture (10 μL) consisted of 20 mM Tris-HCl, pH 7.5, 1 mg/mL of the histones preparation, 0.1 mg/mL (0.67 mM) IgGs was incubated at 37 °C for 2–48 h (standard reaction time was 20 h) as in [42]. The reaction was stopped by adding a mixture containing 50 mM Tris-HCl, pH 6.8, 2% SDS, 10% glycerol, 0.025% bromophenol blue. Histone hydrolysis products were analyzed in 15% SDS-PAGE. After staining of gels with Coomassie blue R-250 [67] or colloidal silver [66], products of the hydrolysis were registered using Gel Doc XR+ (Bio-Rad, Berkeley, CA, USA). IgGs’ relative proteolytic activity was determined by reducing the intensity of the protein bands of each of the histones determined by Image Lab 6.0 (Bio-Rad, Berkeley, CA, USA). The complete hydrolysis of the substrate is taken as 100%. 

### 4.5. Verification Assay That Antibodies Possess Proteolytic Activity

To prove that antibodies exhibit proteolytic activity, we applied several strict criteria that were developed previously [26,30,31,32,33,34]. IgG preparations were subjected to FPLC gel filtration on a Superdex 200 column under acidic conditions (pH 2.6), leading to the destruction of all non-covalent complexes, as in [42,43,44,45]. An equimolar mixture of nine high-activity schizophrenia IgGs (IgG_mix_) was used for gel filtration after antibodies’ pre-incubation in the same acidic buffer. The obtained fractions were immediately neutralized. The relative proteolytic activity of all fractions was determined as described above. For the analysis of proteolytic activity, 5 μL from each fraction was used.

### 4.6. Analysis of Substrate Specificity of Antibodies

To study substrate specificity of IgGs preparations of two different patients with schizophrenia, they were incubated for 20 h at 37 °C under the same standard conditions (20 mM Tris-HCl pH 7.5, 1 mg/mL of protein-substrates, 0.1 mg/mL IgGs) in the presence five histones, myelin basic protein, human and bovine serum albumin, human milk lactoferrin and human lysozyme. The analysis of hydrolysis products was carried out in 15% SDS-PAGE. The hydrolysis products after staining with Coomassie Blue R-250 were registered as described above.

### 4.7. The Effect of pH on the Proteolytic Activity of IgG

The effect of pH on the activity of IgGs was evaluated using different 50 mM buffers: MES buffer (pH 5.5–6.0), MOPS (pH 6.5–7.5), Tris-HCl (pH 8.0–8.5) Gly-NaOH (pH 9.0–10.5). An equimolar mixture of ten IgG preparations (IgG_mix_) was used for the analysis. The IgGs’ relative proteolytic activity (%), depending on the reaction mixture’s pH, was determined as described above.

### 4.8. Determination of Metal Dependence

The following metal salts were used to analyze the effect of metal ions on the catalytic activity of IgGs: CaCl_2_, MgCl_2_, MnCl_2_, CuCl_2_, CoCl_2_, FeSO_4_, NiSO_4_, ZnSO_4_. An equimolar mixture of ten preparations (IgG_mix_) was dialyzed against 20 mM Tris-HCl (pH 7.5) buffer containing 50 mM EDTA to absorb metal ions. Then, the proteolytic activity of IgGas was determined before and after its dialysis against EDTA and after the addition of external metal ions to the reaction mixtures at fixed 2 mM concentration. The IgGs relative proteolytic activity (%) was determined as described above.

### 4.9. Kinetic Parameter Analysis

The study of kinetic parameters was carried out in 20 mM Tris-HCl (pH 7.5) at a fixed concentration of IgG—0.1 mg/mL (0.67 mm) and the following concentrations of histones: 0.2, 0.3, 0.4, 0.5, 0.8, 1, 1.5 mg/mL, as in [42,43,44,45]. The products of the histones’ hydrolysis were analyzed using SDS-PAGE as described above. The specific activity was calculated using the Image Quant program by the substrate loss due to hydrolysis. The kinetic constants were then determined using non-linear approximation by the least square method in the Origin 9.0 program (OriginLab Corporation, Northampton, MA USA); kinetic data were presented in the Lineweaver–Burk coordinates [68].

### 4.10. Statistical Analysis of the Data

Statistical analysis was carried out in the STATISTICA 10 program (StatSoft, USA). To verify the normality of the law of distribution of the obtained data, the Shapiro–Wilk W-test was used. Most of the series of compared parameters did not correspond to the normal distribution law. Therefore, the Mann–Whitney U test and the Kruskal–Wallis ANOVA test were used to assess differences between the samples. Differences between the studied groups were considered statistically significant if *p* < 0.05. Data are presented as mean and standard deviation, as well as median (M) and quartiles (Q1; Q3). For the correlation analysis, the non-parametric Spearman rank method was applied.

## 5. Conclusions

There are certain limitations in this work that could affect the interpretation of the research results. It is generally accepted that Magnetic Resonance Imaging (MRI), electroencephalogram (EEG) and lumbar puncture are methods to exclude an organic cause of psychosis [69,70]. However, the use of these methods in our study had some limitations. Individuals from the control group did not undergo brain MRI, EEG or lumbar puncture just before the start of the study, because they did not demonstrate indications for such examinations. Since all patients included in the study were readmitted to the clinic, they did not undergo a highly invasive lumbar puncture procedure. They also did not undergo MRI brain scans due to the high risk of anxiety-related reactions.

Schizophrenia is associated with chronic low-grade inflammation [12,13,14,15]. It can be initiated by the appearance of cfDNA in the bloodstream, and nucleoproteins, including histones, caused by apoptosis dysfunction [10,11]. However, in addition to inflammation, this can lead to the formation of anti-DNA and anti-histone autoantibodies, among which catalytic antibodies can be found. In this study, we found catalytic antibodies that hydrolytically degrade five histones (H1, H2a, H2b, H3, and H4) in schizophrenia. We have shown that histone hydrolysis is caused by the antibodies themselves, and that the properties of catalytic antibodies differ significantly from canonical proteases, as well as from tight binding autoantibodies. Catalytic antibodies that hydrolyze histones on the one hand can have pathological effects due to catalytic cross-reactivity with MBP. On the other hand, histone-hydrolyzing antibodies can play a compensatory role, since the degradation of histones reduces inflammatory responses. Therefore, they can be considered as a link between humoral immunity and inflammatory reactions.

## Figures and Tables

**Figure 1 ijms-21-07238-f001:**
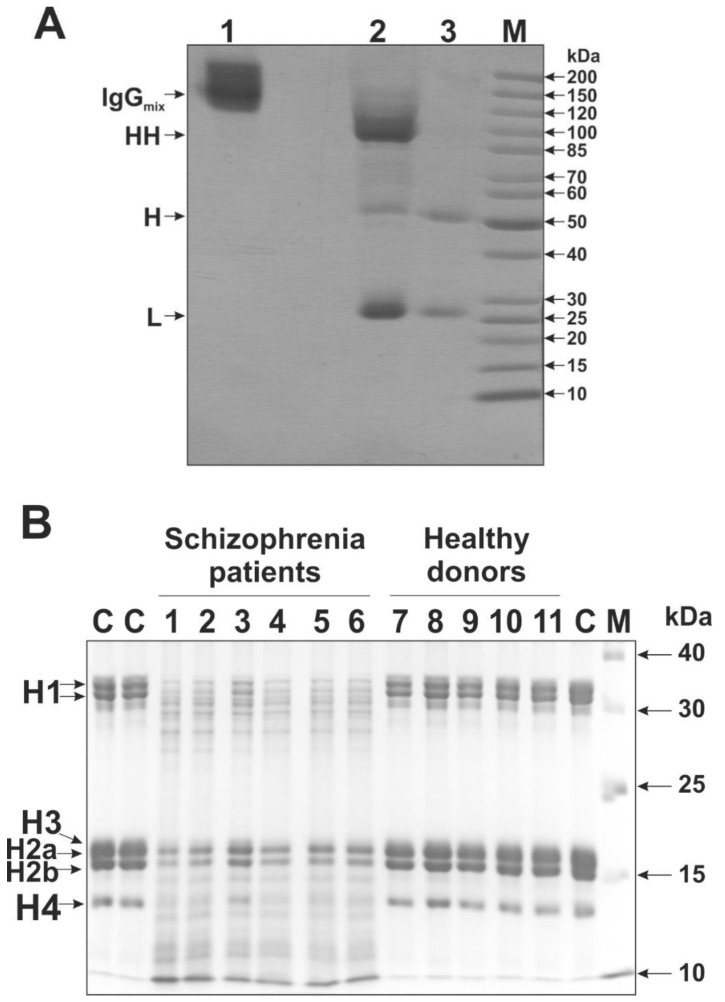
Homogeneity of IgGs isolated from the serum of patients with schizophrenia and analysis of their histone-hydrolyzing activity. (**A**) SDS-PAGE analysis of homogeneity of an equimolar mixture of ten IgG preparations (IgG_mix_) in 4–18% gradient gel followed by silver staining. Lane 1—intact IgG_mix_; lane 2—IgG_mix_ incubated with 40 mM DTT in 37 °C (conditions for the incomplete reduction of disulfide bonds); lane 3—IgG_mix_ incubated with 40 mM DTT at 100 °C (conditions for the complete reduction of disulfide bonds); lane M—protein molecular weight markers. (**B**) SDS PAGE analysis of the proteolytic activity of IgG preparations in the hydrolysis of histones H1, H2a, H2b, H3, H4. The reaction mixtures were incubated for 20 h at 37 °C in the presence of 1 mg/mL histones and 0.1 mg/mL IgGs. Lanes 1–6—IgGs of patients with schizophrenia; lanes 7–11—IgGs of healthy donors; lane C—control reaction mixture without IgG; lane M—protein molecular weight markers. For details, see the Materials and Methods section.

**Figure 2 ijms-21-07238-f002:**
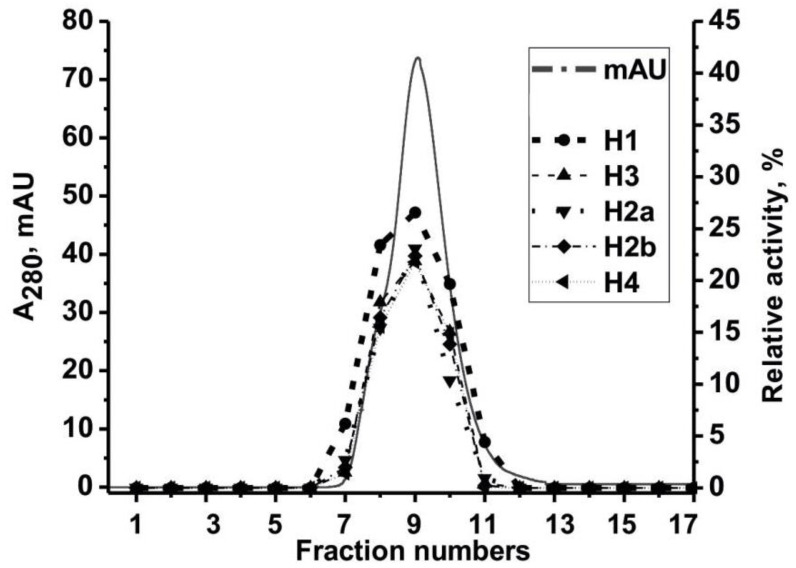
Evidence that histone-hydrolyzing activity belongs to analyzed IgGs of schizophrenia patients. After FPLC gel filtration of IgG_mix_ on Superdex 200 under conditions of dissociation of non-covalent complexes (10 mM Gly-HCl, pH 2.6), the relative activity (%) of the obtained fractions in the hydrolysis of histones was determined: (—), absorbance at 280 nm (A_280_); the designation of hydrolyzable histones is shown in the Figure. Complete hydrolysis of each of five histones for 20 h in the presence of 5 μL of each eluate was taken for 100%. For any details, see the Materials and Methods section.

**Figure 3 ijms-21-07238-f003:**
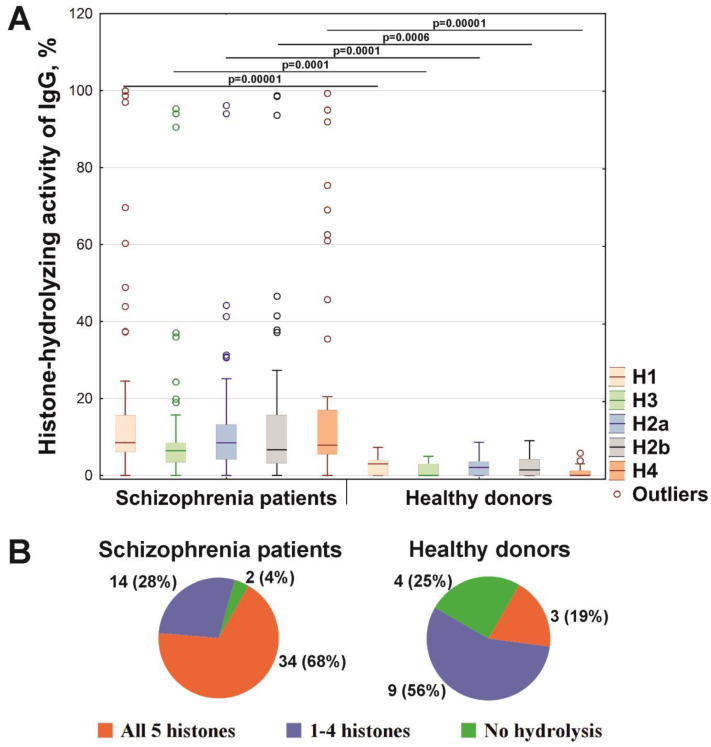
The histone-hydrolyzing activity of IgGs from the sera of patients with schizophrenia and healthy peoples (**A**). The level of histone-hydrolyzing activity of IgGs was normalized to standard conditions (1 mg/mL histone proteins; 0.1 mg/mL IgGs; 20 h of incubation at 37 °C). The complete hydrolysis of the histones as substrates is taken as 100%. Tukey’s test was used to identify outliers. The outliers indicated as open circles. The significance of the differences (*p*) calculated by the Mann–Whitney U test is shown in the Figure. (**B**) Pie chart showing the number and relative percentage of IgG preparations with and without the ability to hydrolyze histones in schizophrenia and healthy donors.

**Figure 4 ijms-21-07238-f004:**
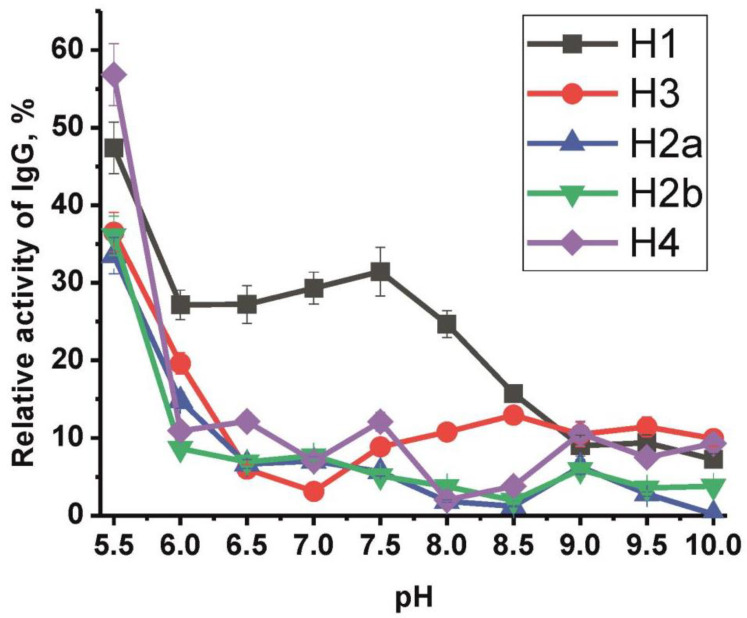
The pH dependence of the relative proteolytic activity of IgG_mix_ in the hydrolysis of five histones. The relative proteolytic activity corresponding to complete hydrolysis of each of the five histones after 20 h of incubation with 0.67 μM IgG_mix_ was taken as 100%. See Materials and Methods for any other details.

**Figure 5 ijms-21-07238-f005:**
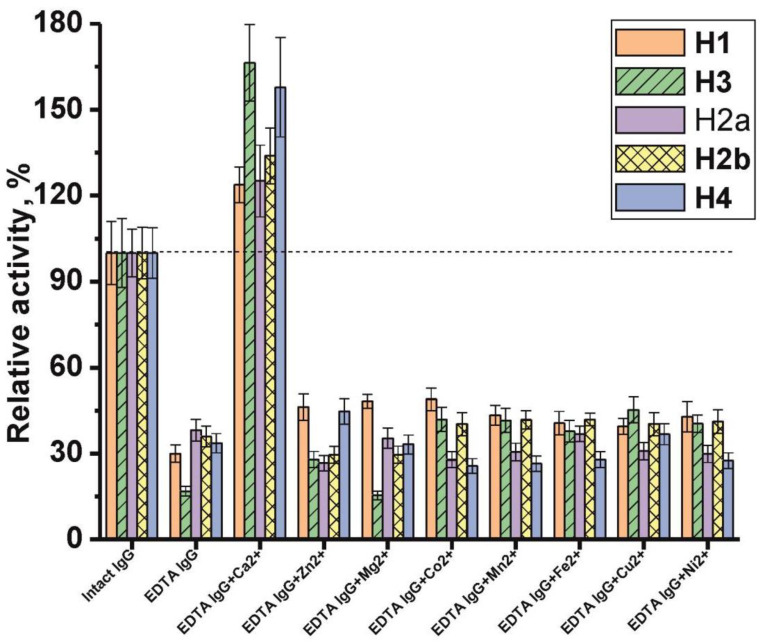
The effect of dialysis of schizophrenia IgG_mix_ against EDTA and the addition of various external metal ions (2 mM) to reaction mixtures in the case of dialyzed IgG_mix_ on the relative activity level in the hydrolysis of five histones. The metal ions used are shown in the Figure. The level of histone hydrolysis by intact IgG_mix_ before dialysis was taken as 100%.

**Figure 6 ijms-21-07238-f006:**
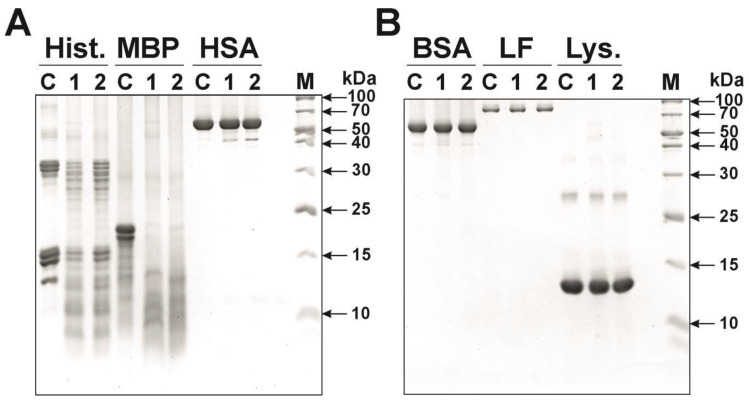
SDS PAGE analysis of substrate specificity of two different IgG-1 and IgG-2 preparations (lanes 1 and 2, respectively) of patients with schizophrenia in the hydrolysis of various proteins: (**A**) five histones (Hist.), myelin basic protein (MBP), human serum albumin (HSA), and (**B**) bovine serum albumin (BSA), human milk lactoferrin (LF), and human lysozyme (Lys.). In all cases, lane C corresponds to various proteins incubated in the absence of IgGs, and lane M corresponds to protein molecular weight markers (**A**,**B**).

**Figure 7 ijms-21-07238-f007:**
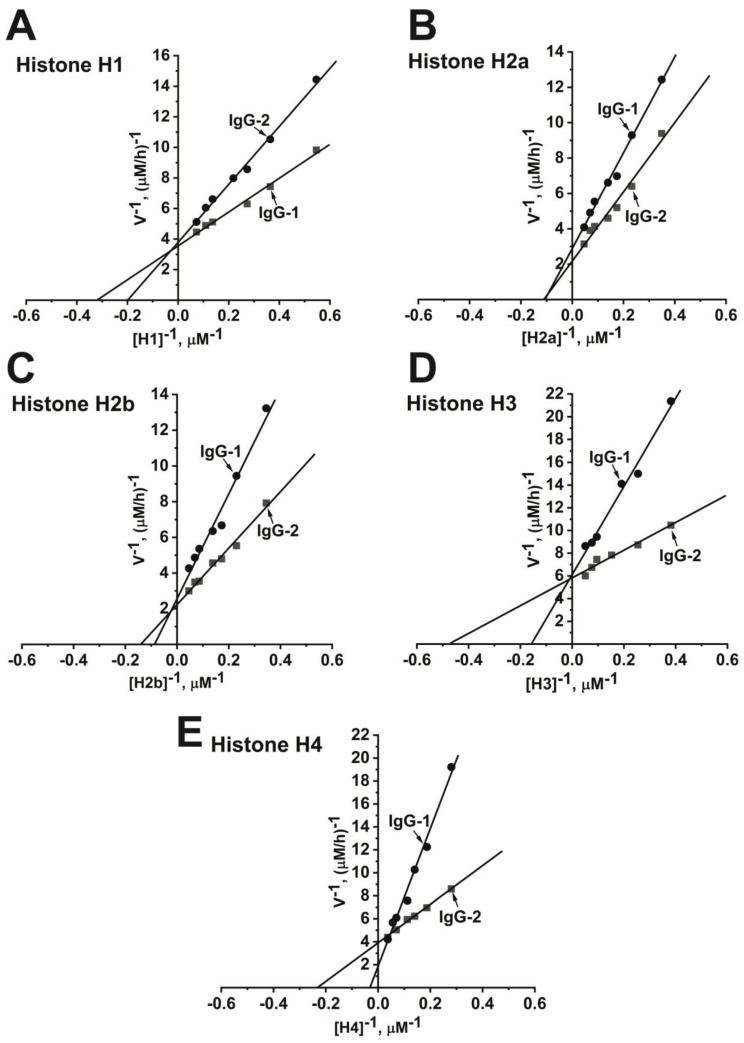
The dependencies of the initial rates of five histones’ ((**A**) H1, (**B**) H2a, (**C**) H2b, (**D**) H3, (**E**) H4) hydrolysis on their concentration in Lineweaver–Burk coordinates in the case of two individual IgG-1 and IgG-2 preparations of two patients with schizophrenia. The error in the initial rate estimation at each substrate concentration did not exceed 10–25%. Reactions were performed as described in Materials and Methods.

**Table 1 ijms-21-07238-t001:** Histone-hydrolysing activity of IgGs depending on leading symptoms of schizophrenia.

Groups of Patients		Histone-Hydrolysing Activity of IgGs, %				
		H1	H2a	H2b	H3	H4
Total group of patients (*n* = 50)	Mean ± SD	18.6 ± 25.3	16.0 ± 22.7	15.7 ± 23.5	12.8 ± 22.0	19.2 ± 26.3
	Median [Q1; Q3]	8.5 [6.1; 15.5]	8.5 [4.3; 13.2]	6.7 [3.3; 15.4]	6.5 [3.6; 8.5]	7.9 [5.7; 16.8]
	Mean ± SD for H1–H4	16.4 ± 23.9				
	Median [Q1; Q3] for H1–H4	7.6 [4.3; 14.4]				
Patients with leading Positive symptoms (*n* = 22)	Mean ± SD	13.4 ± 14.2	12.0 ± 11.0	13.0 ± 9.9	8.6 ± 6.6	17.2 ± 22.0
	Median [Q1; Q3]	7.8 [6.1; 10.7]	10.6 [5.2; 13.0]	9.9 [3.4; 13.5]	7.1 [6.1; 11.3]	7.8 [6.4; 12.8]
	Mean ± SD for H1–H4	12.8 ± 14.3				
	Median [Q1; Q3] for H1–H4	7.8 [5.9; 13.0]				
Patients with leading Negative symptoms (*n* = 28)	Mean ± SD	22.6 ± 31.0	19.2 ± 28.6	18.0 ± 29.3	16.1 ± 28.6	20.8 ± 29.6
	Median [Q1; Q3]	9.3 [6.3; 17.6]	7.3 [4.2; 15.6]	5.6 [3.3; 17.0]	5.3 [3.1; 8.4]	8.8 [5.0; 18.7]
	Mean ± SD for H1–H4	19.3 ± 29.1				
	Median [Q1; Q3] for H1–H4	7.3 [4.0; 17.1]				
Significance of differences (*p*-value) between patients with positive and negative symptoms ^1^		0.374	0.664	0.498	0.341	0.92

^1^*p*-values were calculated using the Mann–Whitney U Test.

**Table 2 ijms-21-07238-t002:** Histone-hydrolysing activity of IgGs depending on schizophrenia types.

Codes of Schizophrenia According to ICD-10 ^1^		Histone-Hydrolysing Activity of IgGs, %				
		H1	H2a	H2b	H3	H4
F20.00 Paranoid schizophrenia with continuous course (*n* = 11)	Mean ± SD	24.3 ± 31.9	21.6 ± 27.5	16.1 ± 27.3	21.6 ± 28.3	23.3 ± 31.2
	Median [Q1; Q3]	11.0 [6.4; 26.4]	12.4 [5.1; 27.8]	5.4 [0; 16.4]	10.6 [5.2; 27.8]	7.8 [6.9; 28.2]
	Mean ± SD for H1–H4	21.4 ± 28.4				
	Median [Q1; Q3] for H1–H4	7.8 [5.1; 27.8]				
F20.01 Paranoid schizophrenia, episodic with progressive deficit (*n* = 14)	Mean ± SD	12.8 ± 13.5	9.3 ± 9.1	9.9 ± 10.4	9.0 ± 4.7	16.6 ± 22.7
	Median [Q1; Q3]	8.4 [6.0; 11.9]	7.0 [2.8; 12.0]	7.8 [4.0; 11.5]	7.4 [6.6; 11.3]	7.6 [4.6; 13.0]
	Mean ± SD for H1–H4	11.5 ± 13.4				
	Median [Q1; Q3] for H1–H4	7.5 [5.1; 12.4]				
F20.02 Paranoid schizophrenia, episodic with stable deficit (*n* = 11)	Mean ± SD	22.7 ± 31.3	17.8 ± 28.0	17.5 ± 30.5	15.6 ± 28.7	21.6 ± 30.7
	Median [Q1; Q3]	7.3 [4.4; 30.9]	6.3 [3.5; 17.9]	3.2 [1.4; 17.3]	3.4 [0; 13.4]	6.5 [2.7; 28.0]
	Mean ± SD for H1–H4	19.0 ± 28.9				
	Median [Q1; Q3] for H1–H4	6.3 [1.7; 21.6]				
F20.09 Paranoid schizophrenia; course uncertain, period of observation too short (*n* = 8)	Mean ± SD	23.1 ± 31.6	22.2 ± 31.0	18.9 ± 32.7	17.0 ± 31.2	22.5 ± 31.4
	Median [Q1; Q3]	9.9 [8.4; 19.3]	10.3 [8.1; 16.3]	7.2 [4.8; 14.8]	7.1 [5.3; 8.1]	11.5 [9.1; 17.4]
	Mean ± SD for H1–H4	20.7 ± 30.0				
	Median [Q1; Q3] for H1–H4	9.2 [6.1; 16.0]				
F20.6 Simple schizophrenia (*n* = 6)	Mean ± SD	7.7 ± 2.7	9.8 ± 3.7	11.6± 6.7	4.9 ± 2.8	9.3 ± 4.9
	Median [Q1; Q3]	7.6 [6.3; 8.8]	10.5 [7.6; 12.9]	10.9 [5.1; 15.6]	5.3 [4.5; 6.0]	7.5 [7.2; 9.2]
	Mean ± SD for H1–H4	8.5 ± 4.6				
	Median [Q1; Q3] for H1–H4	7.6 [5.3; 11.3]				
Significance of differences (*p*-value) between different types of schizophrenia ^2^		0.631	0.661	0.444	0.421	0.684

^1^ ICD-10—International Statistical Classification of Diseases and Related Health Problems. ^2^
*p*-values were calculated using the Kruskal–Wallis ANOVA Test.

**Table 3 ijms-21-07238-t003:** Correlation analysis of histone-hydrolyzing activity of IgGs of schizophrenia patients with clinical data.

Parameters		Histone-Hydrolyzing Activity of IgGs ^1^				
		H1	H3	H2a	H2b	H4
Clinical data	Disease duration	−0.031	−0.016	0.024	0.035	−0.037
	Age	−0.116	0.073	−0.077	−0.094	−0.111
	PANSS Positive ^2^	0.008	0.086	0.130	0.078	0.085
	PANSS Negative ^2^	0.113	−0.084	0.136	0.178	0.190
	Composite index ^2^	−0.077	0.208	0.002	−0.009	−0.060
	PANSS General ^2^	0.248	0.136	0.264	**0.288**	**0.314**
	PANSS Total ^2^	0.115	0.023	0.195	0.177	0.191
Histone-hydrolyzing activity of IgGs	H1	1	**0.728**	**0.847**	**0.667**	**0.839**
	H3	**0.728**	1	**0.670**	**0.624**	**0.703**
	H2a	**0.847**	**0.670**	1	**0.801**	**0.813**
	H2b	**0.667**	**0.624**	**0.801**	1	**0.697**
	H4	**0.839**	**0.703**	**0.813**	**0.697**	1

^1^ Correlation coefficients (*r*) were obtained using the Spearman nonparametric test. Statistically significant (*p* < 0.05) correlation coefficients are indicated in bold. ^2^ Symptom severity of patients with schizophrenia was assessed according to “The positive and negative syndrome scale (PANSS)” [49]. PANSS Positive—positive syndrome scale, which refers to an excess of normal mental functions. PANSS Negative—negative syndrome scale, which represents a diminution or loss of normal mental functions. Composite index—the difference between the PANSS Positive scale scores and PANSS Negative scale scores that reveals the prevailing symptoms. PANSS General—general psychopathology scale, which assesses the general aspects of psychopathology and the severity of schizophrenia. PANSS Total—total points on the PANSS Positive scale, PANSS Negative scale and PANSS General scale.

**Table 4 ijms-21-07238-t004:** Retrospective correlation analysis of histone-hydrolysing activity and nuclease activity of IgGs of schizophrenia patients.

Parameters	Histone-Hydrolysing Activity of IgGs ^1^				
	H1	H3	H2a	H2b	H4
DNAse activity of IgG ^2^	**0.467**	−0.002	**0.478**	0.256	**0.558**
RNAse activity of IgG in the hydrolysis of miR-219a-5p ^3^	**0.542**	0.120	0.273	0.221	**0.506**
RNAse activity of IgG in the hydrolysis of miR-219a-2-3p ^3^	**0.474**	0.089	0.214	0.190	**0.453**
RNAse activity of IgG in the hydrolysis of miR-9-5p ^3^	0.377	0.058	0.088	0.110	0.367
RNAse activity of IgG in the hydrolysis of miR-137 ^3^	−0.352	−0.161	−0.303	−0.293	−0.217

^1^ Correlation coefficients (*r*) were obtained using the Spearman nonparametric test. Statistically significant (*p* < 0.05) correlation coefficients are indicated in bold. ^2^ Data were obtained on the same samples of patients and published previously [26]. ^3^ Data were obtained on the same samples of patients and published previously [35,36].

**Table 5 ijms-21-07238-t005:** The kinetic parameters characterizing efficiency of histones’ hydrolysis by two individual IgG preparations of patients with paranoid schizophrenia.

Histones as Substrates	Kinetics Parameters	IgG-1 ^1^	IgG-2 ^1^
H1	*K*_m_, μM	2.8 ± 0.4	4.8 ± 0.2
	*k*_cat_, min^−1^	(6.8 ± 0.2) × 10^−3^	(6.5 ± 0.1) × 10^−3^
H2a	*K*_m_, μM	7.1 ± 1.1	8.9 ± 1.0
	*k*_cat_, min^−1^	(10.2 ± 0.7) × 10^−3^	(8.4 ± 0.4) × 10^−3^
H2b	*K*_m_, μM	6.4 ± 0.6	8.0 ± 1.1
	*k*_cat_, min^−1^	(10.7 ± 0.4) × 10^−3^	(8.0 ± 0.5) × 10^−3^
H3	*K*_m_, μM	2.3 ± 0.4	5.6 ± 0.8
	*k*_cat_, min^−1^	(4.4 ± 0.2) × 10^−3^	(3.9 ± 0.2) × 10^−3^
H4	*K*_m_, μM	4.7 ± 0.3	24.1 ± 4.2
	*k*_cat_, min^−1^	(6.6 ± 0.1) × 10^−3^	(11.1 ± 1.2) × 10^−3^

^1^ The average values are reported as mean ± SD; at least three measurements were taken for each value; the measurement inaccuracy of reported values did not exceed 10–25%.

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
