# Peer review of "Natural Catalytic IgGs Hydrolyzing Histones in Schizophrenia: Are They the Link between Humoral Immunity and Inflammation?"

_ijms, 2020, doi:10.3390/ijms21197238_

Round 1

Reviewer 1 Report

Dear Authors.

1) Is there any special reason for using ICD10 instead of ICD11?

2) We would also like to know how many of your subjects (controls and schizophrenia patients) did:

a) Brain MRI to exclude CNS tumors?

b) EEG to exclude temporal lobe epilepsy?

c) Lumbar Puncture to exclude encephalitis?

d) Blood sample analysis to exlude autoimune disorders?

e) Urine sample to exclude drug abuse?

Whitout that exams it is not possible to exclude secure "organic" cause of psychosis with certainty (eg secondary schizophrenia).

Good luck with future enterprises. Best regards.

Author Response

The authors are grateful to Reviewer 1 for a critical analysis of our work. Criticism enhances the quality of the manuscript, as well as contributes to the professional growth of the researcher and improves his/her skills for future projects. Based on your comments, we have made some changes to the manuscript. We marked all manuscript adjustments in red.

Please note that your review comments are shown in italic below and our replies in non-italic.

On the comment of [1) Is there any special reason for using ICD10 instead of ICD11?]

Reply: The new edition of International Statistical Classification of Diseases and Related Health Problems (ICD-11), presented to the World Health Assembly in 2019, will enter into force on January 1, 2022. Currently, there is no World Health Organization (WHO)-approved adaptation of ICD-11 in the Russian Federation, so it cannot be used in clinical practice. In this regard, in this work to establish the diagnosis of patients we used ICD-10, which at the time of the work was the most relevant version approved by WHO.

On the comment of [2) We would also like to know how many of your subjects (controls and schizophrenia patients) did:

  1. a) Brain MRI to exclude CNS tumors?
  2. b) EEG to exclude temporal lobe epilepsy?
  3. c) Lumbar Puncture to exclude encephalitis?
  4. d) Blood sample analysis to exlude autoimune disorders?
  5. e) Urine sample to exclude drug abuse?

Whitout that exams it is not possible to exclude secure "organic" cause of psychosis with certainty (eg secondary schizophrenia).]

Reply: We agree that we did not accurately describe the patients’ characteristics in our manuscript. Indeed, psychosis can be caused by various reasons, including “organic” ones. In addition, it is necessary to exclude autoimmune diseases and substance use in patients. Therefore, it is necessary to very clearly describe the characteristics of the patients included in the study, as well as the criteria for inclusion and exclusion in the study. To exclude discrepancies in the interpretation of patients’ diagnoses, we described in more detail these criteria for inclusion and exclusion in the study and made the appropriate changes in the manuscript (please see lines 133-135 and 515-526). Inclusion criteria were the following: paranoid or simple schizophrenia according to the International Statistical Classification of Diseases and Related Health Problems, 10th Revision (ICD-10: F20.0 and F20.6). Exclusion criteria for all study participants were the presence of any chronic infections, inflammatory and autoimmune diseases, and acute viral or bacterial infections no less than two months before the investigation, neurological diseases, other organic mental disorders, mental retardation, and drug addiction. Exclusion criteria for the control group were the presence of acute and chronic infectious, inflammatory, autoimmune, or neurological diseases, and organic brain disorders as well as episodes of drug use. Thus, participants with symptoms of active inflammatory or autoimmune diseases, epilepsy, encephalitis that contribute to the development of organic psychosis were not included in the study.

It should be noted that the selection of patients for our study is carried out by psychiatrists who have international certificates and who are professionals. Therefore, when making a diagnosis, they excluded all possible causes of psychosis, including CNS tumors, temporal lobe epilepsy, encephalitis, autoimmune disorders, and drug abuse. In addition, the patients were monitored for a long time which also increases the accuracy of diagnosis. However, according to the recommendations of the Ministry of Health of the Russian Federation, MRI, EEG are not included in the list of medical services for patients diagnosed with F20 schizophrenia. Patients with signs of "organic" causes of psychosis undergo the necessary diagnostic procedures, and receive a different prevailing diagnosis, and therefore cannot be included in our study.

Reviewer 2 Report

The authors showed that the anti-histone IgG in patients with schizophrenia have a hydrolyzing effect on histone.

Their findings are very interesting and can be a biomarker of schizophrenia, but the description of the theoretical basis of why such a phenomenon occurs is poor.

What is the reason why such a phenomenon does not occur with antibodies in healthy people? Is it because anti-histone antibodies do not exist in the healthy controls? Then the authors should show what extent this antibody is present in patients with schizophrenia. It should not be 100%.

If anti-histone antibodies exist both in healthy subjects and patients with schizophrenia, it should be shown how could it be possible that only antibodies in schizophrenia have a hydrolyzing effect. If the anti-histone IgGs themselves are the same between control and schizophrenia, both of them should have hydrolyzing effects.

Although authors’ theory and experimental results are interesting, it is necessary to show the theoretical rational of existence of such antibodies with convincing experimental data or discussion.

The change of the title may be necessary because it is to speculative in the present form.

Author Response

The authors are deeply grateful to Reviewer 2 for a thorough analysis of our manuscript. Your questions allowed us to look at our work from a different angle and present it more clearly. We made some changes to the manuscript and marked all text adjustments in red.

Please note that your review comments are shown in italic below and our replies in non-italic.

On the comment of [The authors showed that the anti-histone IgG in patients with schizophrenia have a hydrolyzing effect on histone.

Their findings are very interesting and can be a biomarker of schizophrenia, but the description of the theoretical basis of why such a phenomenon occurs is poor.

What is the reason why such a phenomenon does not occur with antibodies in healthy people? Is it because anti-histone antibodies do not exist in the healthy controls? Then the authors should show what extent this antibody is present in patients with schizophrenia. It should not be 100%.

If anti-histone antibodies exist both in healthy subjects and patients with schizophrenia, it should be shown how could it be possible that only antibodies in schizophrenia have a hydrolyzing effect. If the anti-histone IgGs themselves are the same between control and schizophrenia, both of them should have hydrolyzing effects.

Although authors’ theory and experimental results are interesting, it is necessary to show the theoretical rational of existence of such antibodies with convincing experimental data or discussion.]

Reply: The generation of antibodies with catalytic activity is associated with the formation of autoantibodies. In schizophrenia, dysfunction of apoptotic cell death is found, which leads to the release of cell-free DNA (cfDNA), and also nucleoproteins including histones into the bloodstream [1, 2]. This can lead not only to inflammatory responses after recognition of cfDNA and histones by Toll-like receptors [2], but also to their presentation as antigens and the generation of autoantibodies. This assumption is supported by the fact that there is an increase in anti-DNA antibody titers in schizophrenia [3, 4]. Anti-DNA antibodies are also known to cross-bind to the nucleosome and histone proteins [5, 6]. It can be assumed that IgGs with histone-hydrolyzing activity are present among generated anti-DNA and anti-histone autoantibodies. Thus, one of the reasons for the generation of histone-hydrolyzing IgGs in schizophrenia may be an increased level of cfDNA and histones in the bloodstream due to dysfunction of apoptosis. We have added this information to the manuscript (please see lines 57-63 and 365-375).

In our work, we found that the histone-hydrolyzing activity level of IgGs from the serum of schizophrenic patients was 6.1–20.2-fold higher compared to the IgGs activity of healthy donors. IgGs from healthy volunteers had low activity. However, it should be noted that only 20% of IgGs of patients with schizophrenia had high activity (> 20% substrate hydrolysis). These data indicate that these patients have more anti-histone antibodies. These data are consistent with the available data that 23% of schizophrenic patients have anti-histone IgG, while only 6.6% of healthy subjects were positive for anti-histone IgG [7]. Thus, the level of anti-histone antibodies can be assumed to be associated with catalytic activity. We can say that if there are more anti-histone autoantibodies, then there are more catalytic antibodies among them and, accordingly, the catalytic activity that we register will be higher. Healthy donors have a lower concentration of anti-histone antibodies; therefore, the recorded activity is much lower. A similar example is classical laboratory data on the activity of various enzymes in the blood (for example, ALT, AST, etc.), which reflect the amount of these enzymes in the blood. The release of these enzymes from cells leads to an increase in their level in the blood and an increase in the registered activity, which is a diagnostic sign of diseases. We also have added this information to the manuscript and changed Figure 3 (please see lines 201-203; 210-212; and 378-382, and also Figure 3B).

In addition, we studied the kinetic parameters of the histone hydrolysis reaction influenced by IgG. The Km values reflect the affinity of the enzyme to the substrate (in our case, antibodies to the antigen-substrate). Based on the data obtained, it follows that IgGs of patients with schizophrenia have a higher affinity than canonical proteases. However, it should be noted that pathogenic tight binding autoantibodies usually have a high affinity (Kd range: 10-6–10-10 M) [8]. Therefore, some antibodies in schizophrenia acquire catalytic activity and rather high rates of catalysis due to their lower affinity than that of tight binding autoantibodies. Binding affinity is determined by the number of bonds formed with the antigen and accordingly, is determined by the amino acid residues that form the CDR [9]. Therefore, it can be assumed that the antibodies of patients with catalytic activity have a specific CDR structure. However, how the characteristic motifs (catalytic triad) of typical classical proteases are incorporated into the CDRs of antibodies, how their affinity matures, and which specific B-lymphocytes produce catalytic antibodies are not yet fully understood. Further research should be directed toward investigating this issue. We also have added this information to the manuscript (please see lines 324-327 and 451-459).

References:

  1. Jiang, J., Chen, X., Sun, L., Qing, Y., Yang, X., Hu, X., ... & He, L. (2018). Analysis of the concentrations and size distributions of cell-free DNA in schizophrenia using fluorescence correlation spectroscopy. Translational psychiatry, 8(1), 1-8.
  2. Ershova, E. S., Jestkova, E. M., Martynov, A. V., Shmarina, G. V., Umriukhin, P. E., Bravve, L. V., ... & Bogush, M. (2019). Accumulation of circulating cell-free CpG-enriched ribosomal DNA fragments on the background of high endonuclease activity of blood plasma in schizophrenic patients. International journal of genomics, 2019.
  3. Sirota P. Firer, M.A.; Schild, K.; Tanay, A.; Elizur, A.; Meytes, D.; Slor, H. Autoantibodies to DNA in multicase families with schizophrenia. Biol. Psychiatry. 1993, 33, 450-455
  4. 28. Laske, C.; Zank, M.; Klein, R.; Stransky, E.; Batra, A.; Buchkremer, G.; Schott, K. Autoantibody reactivity in serum of patients with major depression, schizophrenia and healthy controls. Psychiatry Res. 2008, 158, 83-86
  5. Founel S, Muller S. Antinucleosome antibodies and T-cell response in systemic lupus erythematosus. Ann Med Interne (Paris). 2002; 153: 513-9
  6. Rekvig, O. P. (2015). The anti-DNA antibody: origin and impact, dogmas and controversies. Nature Reviews Rheumatology, 11(9), 530
  7. Chengappa, K. R., Carpenter, A. B., Yang, Z. W., Brar, J. S., Rabin, B. S., & Ganguli, R. (1992). Elevated IgG anti-histone antibodies in a subgroup of medicated schizophrenic patients. Schizophrenia research, 7(1), 49-54.
  8. Eisen, H. N. (2014). Affinity enhancement of antibodies: how low-affinity antibodies produced early in immune responses are followed by high-affinity antibodies later and in memory B-cell responses. Cancer immunology research, 2(5), 381-392
  9. Fukunaga, A., Maeta, S., Reema, B., Nakakido, M., & Tsumoto, K. (2018). Improvement of antibody affinity by introduction of basic amino acid residues into the framework region. Biochemistry and biophysics reports, 15, 81-85

On the comment of [The change of the title may be necessary because it is to speculative in the present form.]

Reply: We have added a clearer description of the relationship between inflammation and the humoral immune system, and have also summarized our ideas in the Conclusion section (see lines 600-612) so, in our opinion, the title now matches the content of the article.

Round 2

Reviewer 1 Report

Dear Authors

I acknowledge all the efforts done by the authors looking for a better manuscript.

But I am afraid that without brain MRI, EEG and LP we cannot exclude secure organic cause for psychosis. I do not believe in schizophrenia diagnosis without those three exams being normal.

I understand that brain MRI, EEG and LP are not widely available in Russia's Psychiatry. They are not either in my own country's psychiatry! Nevertheless we must be rigorous regarding the use of the "schizophrenia" diagnosis.

I would recommend the authors to change the title of the article. Instead of schizophrenia, they should use "schizophrenia like psychosis". Therefore they should also change it thorough the whole manuscript, accordingly.

Otherwise it is not acceptable for publication, in my humble opinion.

Good luck for future works.

Best regards.

Author Response

Dear Reviewer

We would like to express our gratitude to the Reviewer for his/her comments and strict position regarding the issue of the biological foundations in the differential diagnosis of psychosis.

However, we sincerely believe that in this study, schizophrenia reasonably diagnosed and was consistent with the WHO-approved differential diagnosis protocols. In this reply to the highly respected reviewer, we will try to clarify the stages of diagnosis in our study. Perhaps this will help remove some of the contradictions that have arisen.

Undoubtedly, for a correct diagnosis of schizophrenia, it is necessary to exclude possible organic causes of psychosis. The reviewer mentions MRI, EEG and lumbar puncture (LP) as required tests.

The diagnosis was made using structured clinical interviews according to DSM-IV and ICD-10 criteria.

In our practice, MRI is used as a tool for differential diagnosis in the case of the first psychotic episode and allows to exclude a number of serious organic pathologies, such as tumors, ischemic foci, cysts, foci of demyelination, edema, and thus identify an organic mental disorder. We draw your attention to the fact that among the patients included in the study, there were no first-episode patients. The average duration of the disease in the study group was 13.8 ± 8.4 years and the shortest duration of the disease was one year (for one patient). After long-term follow-up and accurate diagnosis, the patients were included in the study after an exacerbation of the disease and subsequent readmission to the clinic.

In our study, for all included patients, an EEG study of the brain was carried out during their stay in the hospital. This allows us to exclude epilepsy as a possible cause of psychosis in these patients.

Since LP is an invasive procedure associated with a number of possible complications, it requires a number of special indications: Suspicion of meningitis, subarachnoid hemorrhage (SAH), nervous system diseases such as Guillain-Barré syndrome and carcinomatous meningitis, therapeutic relief of pseudotumor cerebri, etc. Patients for whom a lumbar puncture was indicated due to their symptoms were not included in our study.

Also, a general blood analysis was performed for all patients, which allowed excluding from the study patients with an acute inflammatory process.

The reviewer suggested replacing the diagnosis of schizophrenia with "schizophrenia like psychosis" in the text of the manuscript. However, the term “schizophrenia-like psychosis” is not psychiatric nosology presented in the ICD 10. The closest nosology " Acute schizophrenia-like psychotic disorder" is applicable if the total duration of the disorder does not exceed one month (F23.2, please see page 86 https://www.who.int/classifications/icd/en/GRNBOOK.pdf ). As we mentioned above, our study included patients with disease duration from 1 to 35 years (median value 12.5 years; Q1 8; Q2 17 years), which makes this diagnosis inapplicable to the examined patients.

Taking into account the comments of the reviewer regarding the clinical verification of the diagnosis, we have added a section Limitations to the Conclusion section of the manuscript (please see lines 615-624). In this section, we indicated that the examined patients did not undergo an MRI brain scan at the start of their readmission. Patients and controls included in the study did not undergo lumbar puncture. Also, individuals from the control group did not undergo EEG examination of the brain.

The authors sincerely thank the Reviewer for a thorough analysis of the clinical characteristics of patients. Of course, the translational value of the results obtained depends on the quality of the clinical data of the patients included in the study and on the homogeneity of the analyzed groups. Your recommendations made it possible to more accurately describe the clinical characteristics of the analyzed patients. We believe that research by many scientists in the field of biomarkers and functional diagnostic methods for schizophrenia will eventually lead to an increase in the accuracy of the diagnosis of schizophrenia, which will not be in doubt.

With best regards,

Authors

Reviewer 2 Report

Thank you for the revised manuscript and I understand the situation.

I think the new manuscript is acceptable. I think the conclusion that anti-histone antibodies in schizophrenia are proteolytic is OK, and I agree that this may be related to inflammation.

However, the title “Natural catalytic IgGs hydrolyzing histones as a link between inflammation and humoral immunity in schizophrenia” is an assumption due to the lack of specific data on link between inflammation and anti-histone antibody. Therefore, the wording is too conclusive as a title. Is it possible to change the title? A possible alternative is “Identification of Natural catalytic IgGs hydrolyzing histones in schizophrenia”.

Author Response

Dear Reviewer

The authors are deeply grateful to the Reviewer for your valuable suggestions. We agree that the title of the article is too speculative, as we did not investigate the specific link between the humoral immune system and inflammation. This link can be seen as our assumption, a call for dialogue. Therefore, we considered your proposal and decided to change the title of the article to "Natural catalytic IgGs hydrolyzing histones in schizophrenia: are they the link between humoral immunity and inflammation?"

Thank you very much for your analysis of our work.

With best regards,

Authors